# Suppression of annexin A1 and its receptor reduces herpes simplex virus 1 lethality in mice

Li-Chiu Wang[1,2☯‡], Shang-Rung Wu[2,3☯‡], Hui-Wen Yao[2☯‡], Pin Ling[2,4], Guey-Chuen Perng[2,4], Yen-Chi Chiu[2,3], Sheng-Min Hsu[5]*, Shun-Hua Chen[2,4]*

**1** School of Medicine, I-Shou University, Kaohsiung, Taiwan, Republic of China, **2** Institute of Basic Medical Sciences, College of Medicine, National Cheng Kung University, Tainan, Taiwan, Republic of China, **3** Institute of Oral Medicine, College of Medicine, National Cheng Kung University, Tainan, Taiwan, Republic of China, **4** Department of Microbiology and Immunology, College of Medicine, National Cheng Kung University, Tainan, Taiwan, Republic of China, **5** Department of Ophthalmology, National Cheng Kung University Hospital, College of Medicine, National Cheng Kung University, Taiwan, Republic of China

☯ These authors contributed equally to this work.
‡ These authors share first authorship on this work.
* shengmin@mail.ncku.edu.tw (S-MH); shunhua@mail.ncku.edu.tw (S-HC)

**Data Availability Statement:** All relevant data are within the manuscript and its Supporting Information files.

## Abstract

Herpes simplex virus 1 (HSV-1)-induced encephalitis is the most common cause of sporadic, fatal encephalitis in humans. HSV-1 has at least 10 different envelope glycoproteins, which can promote virus infection. The ligands for most of the envelope glycoproteins and the significance of these ligands in virus-induced encephalitis remain elusive. Here, we show that glycoprotein E (gE) binds to the cellular protein, annexin A1 (Anx-A1) to enhance infection. Anx-A1 can be detected on the surface of cells permissive for HSV-1 before infection and on virions. Suppression of Anx-A1 or its receptor, formyl peptide receptor 2 (FPR2), on the cell surface and gE or Anx-A1 on HSV-1 envelopes reduced virus binding to cells. Importantly, *Anx-A1* knockout, Anx-A1 knockdown, or treatments with the FPR2 antagonist reduced the mortality and tissue viral loads of infected mice. Our results show that Anx-A1 is a novel enhancing factor of HSV-1 infection. Anx-A1-deficient mice displayed no evident physiology and behavior changes. Hence, targeting Anx-A1 and FPR2 could be a promising prophylaxis or adjuvant therapy to decrease HSV-1 lethality.

## Author summary

Herpes simplex virus 1 (HSV-1)-induced encephalitis is the most devastating consequence of HSV-1 infection, even in patients treated with anti-HSV-1 drugs. Moreover, encephalitis induced by drug-resistant HSV-1 has been reported in immunocompromised patients. Identifying the cellular factors in promoting HSV-1 replication, especially those increasing virus attachment and entry, could facilitate the development of alternative or adjuvant therapy. Here, we identified annexin A1 (Anx-A1) and its receptor, formyl peptide receptor 2 (FPR2), facilitating HSV-1 attachment to the cell surface. Suppression of Anx-A1 or blockage of FPR2 impaired HSV-1 attachment to cells, viral yields in cells, and HSV-1

**Funding:** This work was supported by grants from Taiwan (MOST 108-2320-B-006-035-MY3 and NSC 102-2320-B-006-028-MY3) and funding from the Center of Infectious Disease and Signal Research of National Cheng Kung University (to SHC) and I-Shou University (ISU109-S-02) to LCW. The funders had no role in study design, data collection and analysis, decision to publish, or preparation of the manuscript.

**Competing interests:** The authors have declared that no competing interests exist.

lethality in mice. Moreover, blocking FPR2 decreased the replication of drug-resistant HSV-1 in BABL/c nude mice. Hence, targeting Anx-A1 and FPR2 could be alternative or adjuvant therapy for HSV-1 infection.

## Introduction

Herpes simplex virus (HSV)-induced encephalitis is the most common cause of sporadic fatal encephalitis, with an incidence of approximately 1 in 250,000 persons per year [1]. Acyclovir and its derivatives are used for treatment [1,2]. The mortality rates of encephalitis patients without or with treatment are 70% and 30%, respectively [1,2]. Survivors are often left with severe and permanent neurological sequelae, and only 2.5% of encephalitis patients regain normal neurological function [1,3]. There are two species of HSV (HSV-1 and HSV-2), which can infect humans and deposit their genomes in neurons to establish latency. HSV-1 infects the majority (>80%) of adults worldwide [4,5] and accounts for more than 95% of encephalitis cases in patients beyond the neonatal period [6]. During infection, HSV-1 interacts with cellular factors, which can modulate virus infection to affect the severity of virus-induced diseases. Studies are needed to identify these cellular factors and their mechanisms of action for a better understanding of viral pathogenesis and more importantly, for developing strategies to reduce HSV-1 lethality.

During virus entry, viral envelope glycoproteins interact with cellular proteins on the cell surface. HSV-1 has at least ten different envelope glycoproteins [4]. Among these glycoproteins, gB, gD, gH, and gL are essential for virus infectivity. The remaining glycoproteins, including gC, gE, gG, gI, gJ, gK, gM, and gN, can enhance infection but are not absolutely required for virus infectivity [7]. Although gE is nonessential for virus infection of cells, gE deficiency reduces HSV-1 lethality in experimental animals, mice and rabbits [8, 9]. The potential of gE to mediate HSV-1 infection is suggested by the finding that an anti-gE antibody (Ab) neutralizes virus infectivity [10]. Subsequent reports have revealed that the major function of gE is to form a complex with gI to facilitate cell-to-cell spread [9,11]. The receptors of gB and gD have been identified [7,12,13]. However, the ligands for other viral envelope glycoproteins remain elusive, and the significance of these ligands in virus-induced encephalitis is unknown.

One report found that mutation of the cellular *annexin A1* (*Anx-A1*) gene generated by insertion of a retrovirus vector reduces HSV-1 replication in a rat epithelial cell line [14]. Anx-A1, also known as lipocortin-1, calpactin, and p35, belongs to the annexin superfamily, which has 12 members. Annexins are proteins with variable N-termini, that are responsible for their unique biological functions, and a conserved C-terminus, that is consisted of four or eight repeats of 70 amino acids, with calcium-binding activity [15]. Anx V, which serves as an apoptosis marker, is a famous member in the superfamily [15]. Anx-A2 fails to affect HSV-1 replication in cells [14]. Anx-A1 is abundantly expressed in epithelial cells and neutrophils. It is mainly in the cytoplasm and can be secreted and bind to negatively charged phospholipids or proteins on the cell surface via calcium [15,16]. Anx-A1 binds to its receptors, formyl peptide receptor (FPR) 1 and 2, to regulate inflammatory responses and many cellular activities, such as exocytosis, endocytosis, and migration [15], but its deficiency does not result in apparent defects in mice [17]. Very few studies have investigated how Anx-A1 enhances HSV-1 replication, and the present report was designated to address this issue. Our in vitro results showed that Anx-A1 interacts with gE to facilitate HSV-1 infection of cells. Moreover, blocking FPR2,

but not FPR1, decreases HSV-1 binding to cells. In vivo results revealed that suppression of Anx-A1 or FPR2 reduces the mortality and tissue viral loads of infected mice.

## Results

### Anx-A1 can be detected on the surface of cells permissive for HSV-1 before infection

Anx-A1 is a cytosolic protein and can be externalized and detected on the cell surface [15]. To examine the expression of Anx-A1 before HSV-1 infection in cells that have been shown to express Anx-A1 and support virus replication, we used the human epithelial cell line, A549, as the cell type is the major component of mucosa and skin, which are primary HSV-1 entry and replication sites. Anx-A1 on the cell surface of mock-infected A549 cells was extracted by treating cells with the calcium chelator, EDTA, because Anx-A1 attaches to the cell membrane via calcium [15]. Afterward, the EDTA-treated cells were harvested and analyzed for Anx-A1 in the cytoplasm. In mock-infected cells, Anx-A1 was detected on the cell surface at a level less than that in the cytoplasm as determined by western blotting (S1A Fig). Similar results were obtained when another cell line (N18), which is derived from mouse neuroblastoma and permissive for HSV-1 replication [18], was used for the assay (S1B Fig). These results demonstrate the expression of Anx-A1 on the surface of cells permissive for HSV-1 before infection in a manner not specific to human or mouse cells and a particular cell type. We also performed immunofluorescence staining to detect Anx-A1 on the cell surface. Because Anx-A1 can be detected on the cell surface and in the cytoplasm, we first evaluated our assay and confirmed that staining of cells before permeabilization failed to detect the intracellular protein, F-actin (S2 Fig). Using this assay, we stained mock-infected A549 cells using anti-Anx-A1 Ab before permeabilization and detected Anx-A1 on the surface (Fig 1) of 32% cells.

### Cell surface Anx-A1 promotes HSV-1 attachment

To study the role of Anx-A1 in HSV-1 infection, we determined the viral titers in primary embryonic fibroblasts cultured from mice (MEFs) infected with virus at the multiplicity of infection (MOI) of 0.01. The mean viral titers in *Anx-A1$^{-/-}$* MEFs at 24 to 48 h postinfection (hpi) were lower than those of WT MEFs (Fig 2A). We reconstituted Anx-A1 expression in *Anx-A1$^{-/-}$* MEFs by transfecting cells with the plasmid carrying Anx-A1 with a 6×His tag (S3 Fig). The mean viral yields were increased in *Anx-A1$^{-/-}$* MEFs with ectopic Anx-A1 expression from 24 to 48 hpi (Fig 2B), showing that Anx-A1 enhances HSV-1 production.

We detected Anx-A1 on the surface of cells permissive for HSV-1 before infection. Treatment with recombinant Anx-A1 has been shown to reduce cytomegalovirus infection by an

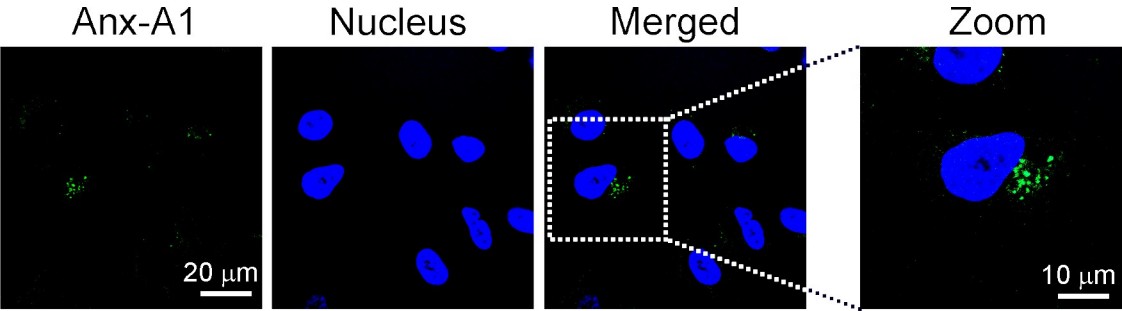

**Fig 1. Anx-A1 can be detected on the surface of cells permissive for HSV-1 before infection.** Mock-infected A549 cells were subjected to immunofluorescence staining with anti-Anx-A1 Ab before permeabilization. Nuclei were counterstained with Hoechst.

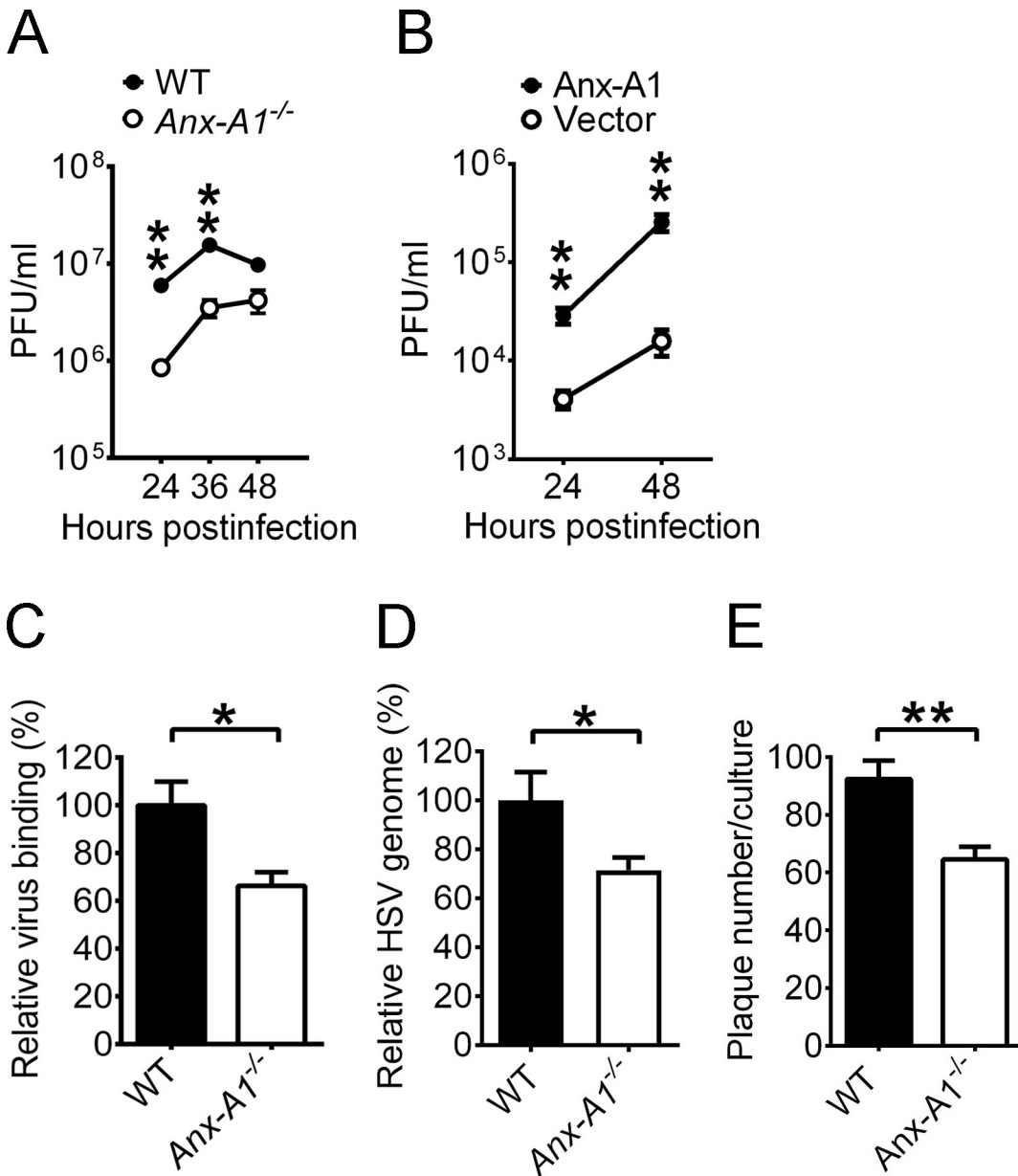

**Fig 2. Cell surface Anx-A1 promotes HSV-1 attachment.** (**A**) The HSV-1 yields in WT or *Anx-A1*⁻/⁻ MEFs (MOI = 0.01) at the indicated times are shown. (**B**) The viral yields in infected *Anx-A1*⁻/⁻ MEFs (MOI = 0.01) transfected with a control vector (Vector) or the vector expressing His-tagged Anx-A1 (Anx-A1) at the indicated times are shown. (**C and D**) The levels of HSV-1 binding on MEFs determined by plaque assay (**C**) or by real-time PCR (**D**) are shown. The level of HSV-1 binding to WT MEFs was set as 100%. (**E**) The numbers of HSV-1 plaques formed on MEFs monolayers after infection with 400 PFU/ monolayers of virus are shown. Data show the mean + or ± SEM of >4 samples per group. *$P < 0.05$ and **$P < 0.01$.

unknown mechanism in human foreskin fibroblasts [19], suggesting the potential of cell surface Anx-A1 to mediate virus binding. We studied the interaction between HSV-1 and Anx-A1 and observed colocalization of HSV-1 and Anx-A1 on the cell surface using immunofluorescence staining (S4 Fig). To investigate whether Anx-A1 affects HSV-1 attachment, MEFs were infected with HSV-1 for 1 h at 4˚C, washed at 4˚C, immediately stored at -80˚C, and subjected to determine the numbers of virus binding via plaque assay as described [20] or via real-

time PCR. Using either plaque assay (Fig 2C) or real-time PCR (Fig 2D), the levels of virus attached to the cell surface were reduced in *Anx-A1⁻/⁻* MEFs by about 30%. We further determined whether Anx-A1-enhanced attachment increases the number of plaques formed on MEF monolayers. Cells were infected with HSV-1 at room temperature for 1 h, washed with acidic citric buffer (pH = 3) to inactivate the remaining virus on the cell surface, and incubated in the medium with methylcellulose to form plaques. The number of HSV-1 binding and penetrating into *Anx-A1⁻/⁻* MEFs was decreased by 30% (Fig 2E). Taken together, the results demonstrate that Anx-A1 on the cell surface enhances HSV-1 attachment to cells.

## Anx-A1 interacts with HSV-1 gE

We further validated the binding of HSV-1 to Anx-A1 using a virus overlay protein binding assay. The recombinant, His-tagged Anx-A1 and the control protein (bovine serum albumin, BSA) were separated, blotted on membrane, and incubated with purified HSV-1 virions, and the binding of virus to protein was detected by anti-HSV-1 Ab. Fig 3A shows the binding of HSV-1 to Anx-A1, but not to BSA.

We next identified the HSV-1 envelope glycoprotein(s) interacting with Anx-A1 in the lysate of infected A549 cells using anti-Anx-A1 Ab for immunoprecipitation followed by LC-MS/MS analysis. The results revealed that gE, gI, and gB interacted with Anx-A1 with Mascot scores of 810, 119, and 62, respectively, suggesting a high probability of binding between gE and Anx-A1 (S1 Table). To confirm the LC-MS/MS results, we assayed A549 cell lysates. Anx-A1 was detected in lysates of mock-infected and infected cells, while gB, gD, and gE were detected in the lysate of infected cells, but not in that of mock-infected cells (Fig 3B top left panel). The gD, unable to bind Anx-A1, was served as a negative control. Anti-Anx-A1 Ab, but not control Ab, precipitated Anx-A1 as demonstrated by immunoprecipitation of lysates followed by immunoblotting analysis (Fig 3B top right panel). More importantly, gE, but not gB or gD, was co-precipitated with Anx-A1 from the lysate of infected cells. The lysates were also immunoprecipitated with anti-gE or control Ab followed by immunoblotting analysis (Fig 3B bottom right panel). The anti-gE Ab reacted strongly with gE, and Anx-A1 was co-precipitated with gE. The immunoprecipitation results show the interaction of Anx-A1 with gE, but not with gB or gD. HSV-1 gE and gI form a complex [9,21]. To study whether the interaction of gE with Anx-A1 requires other viral proteins, we infected the human fibroblast cell line (293) with or without the recombinant adenovirus expressing HSV-1 gE. Cells infected with HSV-1 served as a positive control. In the lysates of adenovirus- or HSV-1-infected cells, gE was detected (Fig 3C). Anx-A1 was co-precipitated with gE from lysates of cells infected with adenovirus or HSV-1 with the Anx-A1/gE ratios of 0.44 and 0.51, respectively, showing that the interaction of gE with Anx-A1 does not require other HSV-1 proteins.

We mapped the domains of Anx-A1 and gE needed for the interaction. HSV-1 gE contains four domains [22], a signal peptide containing amino acids (a.a.) 1–25, an extracellular domain (26–421 a.a.), a transmembrane domain (422–447 a.a.), and a short cytoplasmic tail (448–550 a.a.). Lysates of 293 cells overexpressing different lengths of Flag-tagged gE were immunoprecipitated with control or anti-Anx-A1 Ab (Fig 3D). The full-length gE (1–550 a.a.) and gE without the cytoplasmic tail (Δ447–550) were co-precipitated with Anx-A1. However, gE without the extracellular domain (Δ1–421) failed to be co-precipitated with Anx-A1. Anx-A1 consists of two domains [15], an N-terminus (1–42 a.a.) and a core domain responsible for calcium binding (43–346 a.a.). Lysates of 293 cells overexpressing different lengths of His-tagged Anx-A1 were mock-infected or infected with HSV-1 (Fig 3E). Both full-length Anx-A1 and Anx-A1 lacking the N-terminus (Δ1–42) were co-precipitated with gE. Collectively, the extracellular domain of gE interacts with the core domain of Anx-A1.

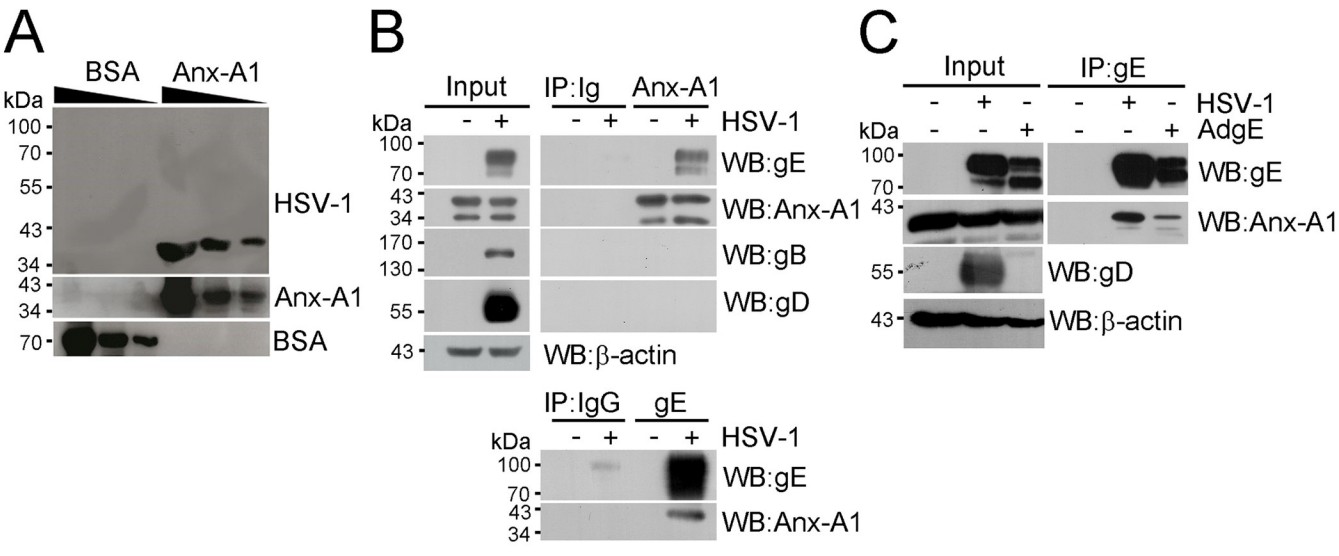

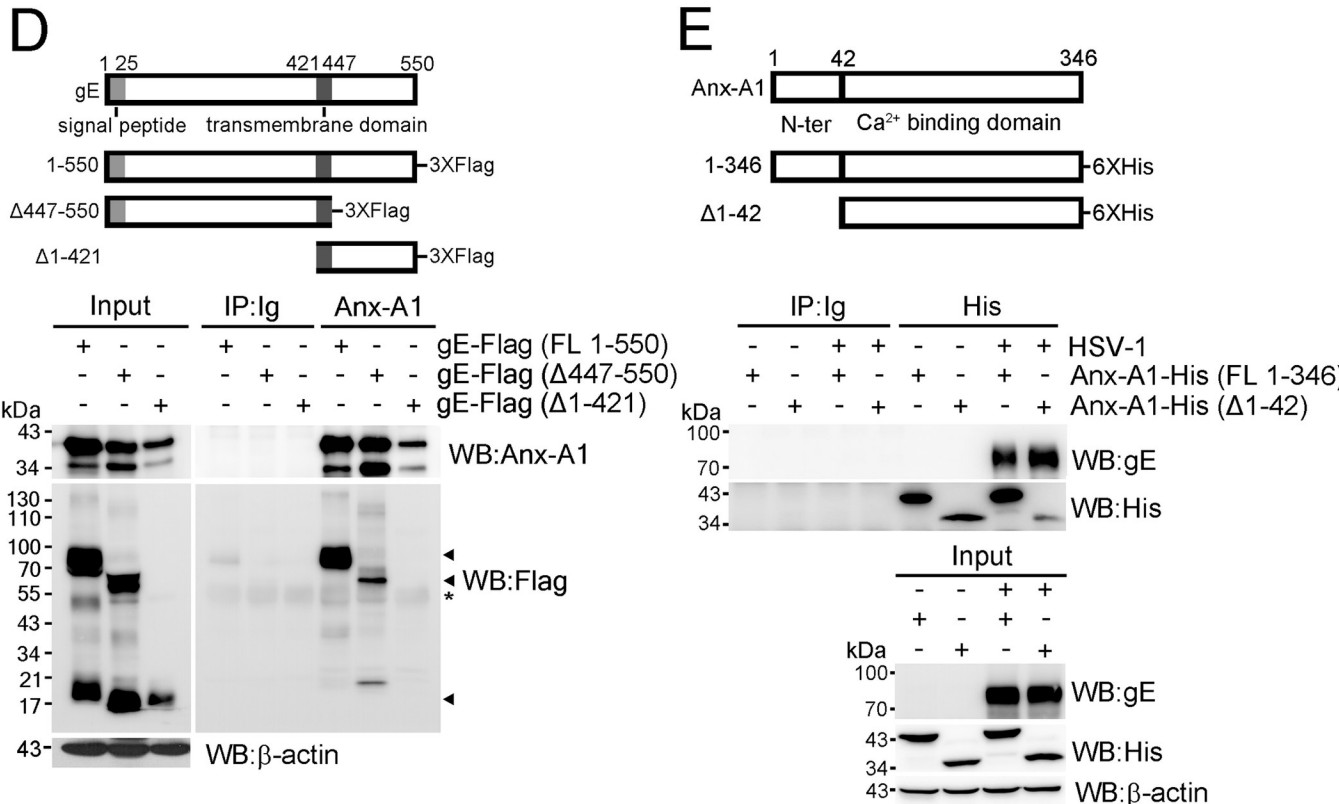

**Fig 3. Anx-A1 interacts with HSV-1 gE.** (**A**) A virus overlay protein binding assay was performed with reducing doses of BSA and His-tagged Anx-A1 (25, 5, and 1 µg) on the membrane before incubation with purified HSV-1 and blotting with the indicated Abs. (**B**) The lysates of A549 cells mock-infected (-) or infected with HSV-1 (+) were subjected to western blotting (Input) or immunoprecipitated (IP) with control Abs (Ig or IgG) or Abs against Anx-A1 or gE before blotting as indicated. (**C**) The lysates of 293 cells mock-infected (-) or infected with HSV-1 (+) or the adenovirus expressing HSV-1 gE (AdgE) were subjected to western blotting (Input) or IP with anti-gE Ab before blotting as indicated. (**D**) A schematic diagram of the Flag-tagged gE deletion constructs is shown in the top panel. The lysates of 293 cells transfected with the indicated Flag-tagged gE constructs were subjected to western blotting (Input) or IP with control or anti-Anx-A1 Ab before blotting as indicated. Black triangles indicate Flag-tagged gE. * indicates heavy chains of Ab. (**E**) A schematic diagram of His-tagged Anx-A1 deletion constructs is shown in the top panel. The lysates of 293 cells transfected with the indicated His-tagged Anx-A1 constructs and mock-infected (-) or infected (+) with HSV-1 were subjected to western blotting (Input) or IP with control or anti-His Ab before blotting as indicated.

## Anx-A1 can be detected on HSV-1 envelopes to enhance virus binding

As Anx-A1 interacts with the gE extracellular domain, it is very likely that Anx-A1 might attach to HSV-1 envelopes. To assess this possibility, we determined the presence of Anx-A1 on HSV-1 virion via transmission electron microscopy. Anx-A1 was co-localized with gE on the HSV-1 envelopes (Fig 4A and 4B). We purified and separated HSV-1 virions into two fractions, one with tegument plus capsid and the other with envelopes. Indeed, Anx-A1 was detected in the envelope fraction but not in the tegument and capsid fraction (Fig 4C). To address the significance of Anx-A1 on virions for virus binding to cells, we propagated virus in WT or $Anx-A1^{-/-}$ MEFs to produce virus with or without Anx-A1 and assessed the virus binding on MEFs. Absence of Anx-A1 on the cell surface or virus envelopes abrogated Anx-A1-enhanced HSV-1 attachment to MEFs (Fig 4D), showing that Anx-A1 on both sides is required for HSV-1 attachment. Similar results were observed in mouse MEFs infected with the HSV-1 gE-null mutant (F-gEβ) and its parental strain (F). Western blot analysis confirmed the absence of gE expression in cells infected with F-gEβ (Fig 4E right panel). The level of HSV-1 F binding to $Anx-A1^{-/-}$ MEFs was reduced when compared with WT MEFs (Fig 4E left panel). More importantly, the level of gE-null mutant binding to WT MEFs was reduced and comparable to that of binding to $Anx-A1^{-/-}$ MEFs, showing that the Anx-A1-mediated enhancement of HSV-1 binding is abolished in the absence of gE.

To neutralize extracellular Anx-A1, MEFs were treated with control or anti-Anx-A1 serum before and during infection with HSV-1. The levels of HSV-1 attachment (Fig 4F) and mean viral yield at 24 hpi (Fig 4G) were lower in MEFs treated with anti-Anx-A1 serum than those with control serum, showing that extracellular Anx-A1 facilitates HSV-1 attachment. Similar results were observed in N18 cells (Fig 4H). Additionally, the N18 cells were treated with different dilutions of control or anti-Anx-A1 serum and infected with HSV-1 strain KOS-GFP (MOI = 10) for 1 h, washed, and harvested to determine the percentage of GFP-positive (HSV-1-infected) cells at 17 hpi. Anti-Anx-A1 serum reduced the percentage of HSV-1-infected cells by 20 to 40% (Fig 4I) in a manner dependent on the Ab concentration. We also tested the human neuronal cell line, SK-N-SH, which is highly susceptible to cold temperature and undergoes morphological changes, making the cells unsuitable for assays at 4°C. Anti-Anx-A1 serum also reduced HSV-1 binding to another human epithelial cell line, U-2 OS (S5 Fig).

## Absence or suppression of Anx-A1 expression reduces the mortality and tissue viral loads of HSV-1-infected mice

We next addressed the significance of Anx-A1 for HSV-1 infection in vivo using a murine model. Mice were infected with virus on the scarified cornea, a site in humans that can be infected by the virus [4,5]. In our infection model, infectious virus is detected in the eye, trigeminal ganglion (TG), and brain, but not in other tissues, organs, or blood, so these three tissues permissive for viral replication were harvested for detection of Anx-A1 by western blotting. In mock-infected WT mice, Anx-A1 was detected in the eye, TG, and brain at the relative levels of 20, 7, and 1, respectively (S6A Fig). Abundant Anx-A1 was detected in the mouse eye, so this tissue was subjected to assessment of the cell type expressing Anx-A1 and whether Anx-A1-expressing cells could be infected by HSV-1 by immunofluorescence staining. In the mock-infected cornea of WT mice, the epithelium positive for keratin K3, the intermediate filament cytoskeleton specifically expressed in corneal epithelial cells [23], was stained by anti-Anx-A1 Ab (S6B Fig), showing that corneal epithelial cells (keratinocytes) express Anx-A1. In the cornea harvested from infected WT mice at 1 days postinfection (dpi), most (>90%) of the HSV-1 antigen-positive cells were also stained by anti-Anx-A1 Ab, showing the infection of Anx-A1-expressing cells by HSV-1 (S6C Fig).

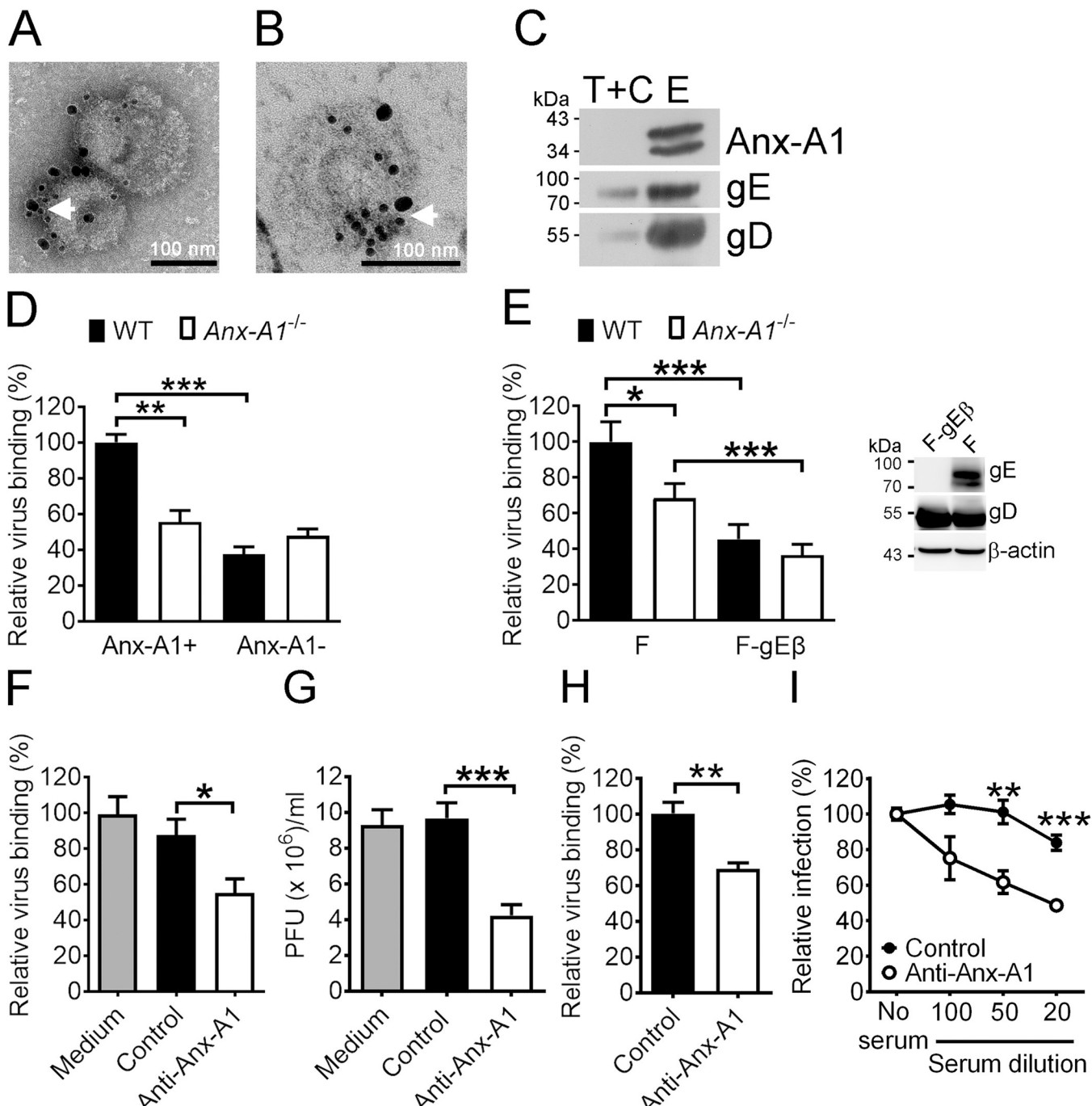

**Fig 4. Anx-A1 on the HSV-1 envelope enhances virus binding to the cell surface.** Representative images of transmission electron microscopy of Anx-A1 and gE on purified HSV-1 virions (**A**) and ultrathin sections of HSV-1 virions (**B**) are shown. The Abs against Anx-A1 or gE were labeled with gold particles of 18 and 10 nm, respectively. The arrows represent the Anx-A1-gE interaction. (**C**) Representative western blots of Anx-A1, gE, and gD in the tegument and capsid (T+C) and envelope (E) fractions of purified HSV-1 are shown. (**D**) The levels Anx-A1-positive (Anx-A1+) or Anx-A1-negative (Anx-A1-) HSV-1 binding on WT or *Anx-A1*[-/-] MEFs are shown. (**E**) The representative western blots of indicated proteins in lysates of Vero cells infected with indicated HSV-1 strains at 16 hpi (right panel) and the levels of strain F or F-gEβ binding to WT or *Anx-A1*[-/-] MEFs (left panel) are shown. The levels of virus binding (**F**) and viral yields at 24 hpi (**G**) of WT MEFs treated without (medium) or with control or anti-Anx-A1 serum (at 1:500 dilution) are shown. (**H**) The levels of HSV-1 binding to N18 cells treated with control or anti-Anx-A1 serum (at 1:100 dilution) are shown. (**I**) The level of GFP-positive N18 cells treated with the indicated dilutions of control or anti-Anx-A1 serum and infected with KOS-GFP (MOI = 10) for 17 h are shown. The levels of Anx-A1-positive HSV-1 binding to WT MEFs or N18 cells treated with control serum and the percentage of GFP-positive N18 cells without treatment were set as 100%. Data show the mean + SEM of >4 per group. *$P < 0.05$, **$P < 0.01$, and ***$P < 0.001$.

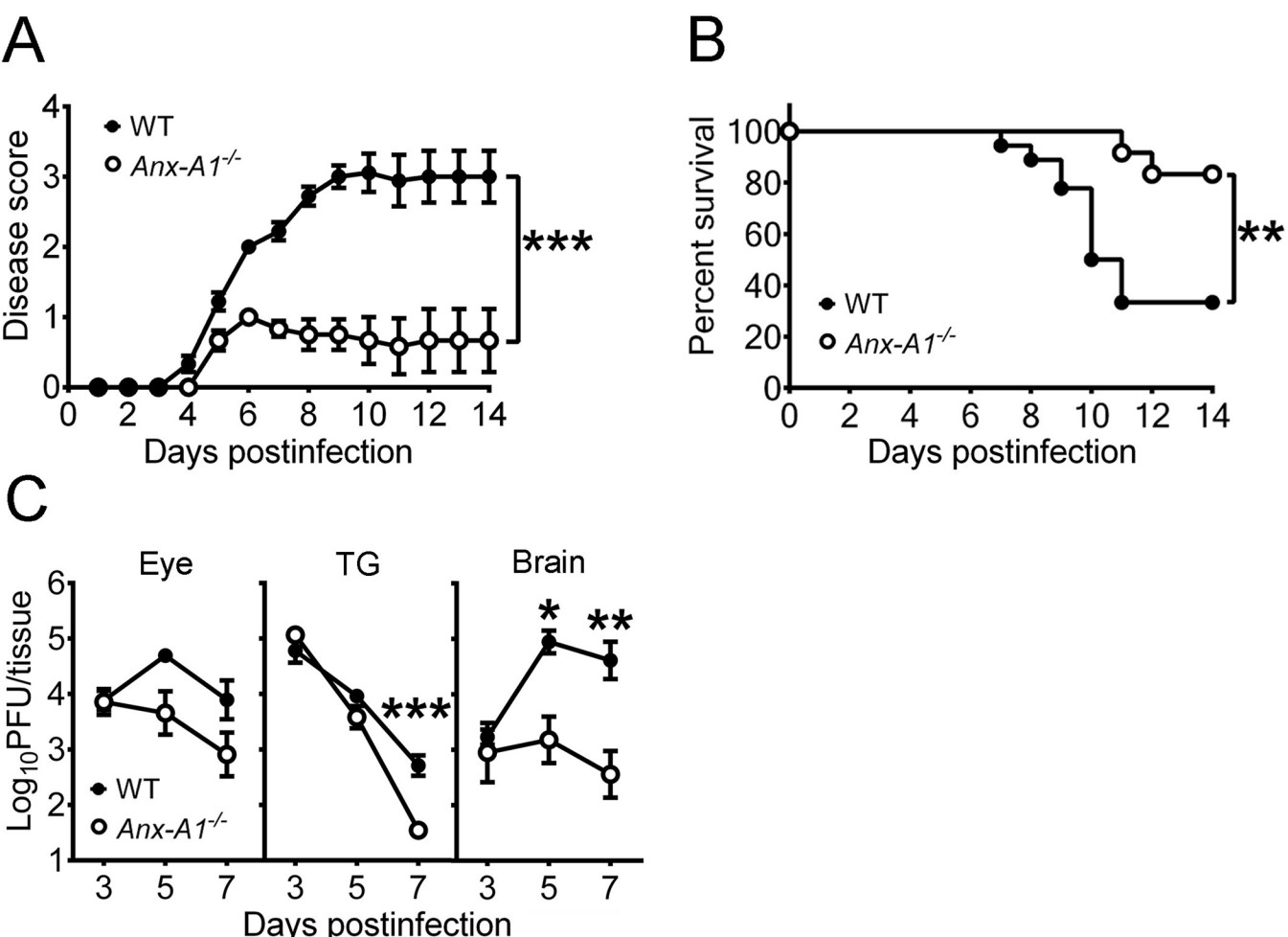

**Fig 5. Absence of Anx-A1 reduces the mortality and tissue viral loads of HSV-1-infected mice.** The disease scores (**A**) and survival rates (**B**) of WT mice ($n = 18$) and $Anx-A1^{-/-}$ mice ($n = 12$) infected with HSV-1 294.1 are shown. (**C**) Viral loads in tissues of infected WT and $Anx-A1^{-/-}$ mice on the indicated dpi are shown ($n > 6$ per data point). Data show the mean ± SEM in panels A and C. $^{*}P < 0.05$, $^{**}P < 0.01$, and $^{***}P < 0.001$.

We compared infected WT and $Anx-A1^{-/-}$ mice for disease progression, survival, and tissue viral loads (Fig 5). Without HSV-1 infection, $Anx-A1^{-/-}$ mice displayed no evident physical and behavioral changes compared with WT mice. After HSV-1 infection, mice, especially WT mice displayed signs of encephalitis after 4 dpi, including ataxia, hunched posture, and lethargy. Approximately two-thirds of WT mice (12/18) displayed signs of severe encephalitis (score > 3) with a mean disease score of 3 at 14 dpi, and one-sixth of $Anx-A1^{-/-}$ mice (2/12) displayed severe encephalitis with a mean disease score of 0.7, which was significantly lower than that of WT mice (Fig 5A). The survival rate of infected $Anx-A1^{-/-}$ mice (10/12) was higher than that of WT mice (6/18) by 14 dpi (Fig 5B). Absence of Anx-A1 reduced the viral loads in brains at 5 and 7 dpi and in TG at 7 dpi (Fig 5C). High levels of mortality rate and brain viral loads were detected in infected WT mice when compared with $Anx-A1^{-/-}$ mice, so we examined the infection of neurons in the mouse brain using immunofluorescence staining to detect NeuN (a neuron-specific protein in the nucleus) and HSV-1 antigen (S7 Fig). In the brains of mock-infected WT and $Anx-A1^{-/-}$ mice, comparable NeuN expression was detected. In the brains of infected mice, more HSV-1-positive neurons were detected in WT mice than in $Anx-A1^{-/-}$ mice at 7 dpi, suggesting that the absence of Anx-A1 protects neurons from infection.

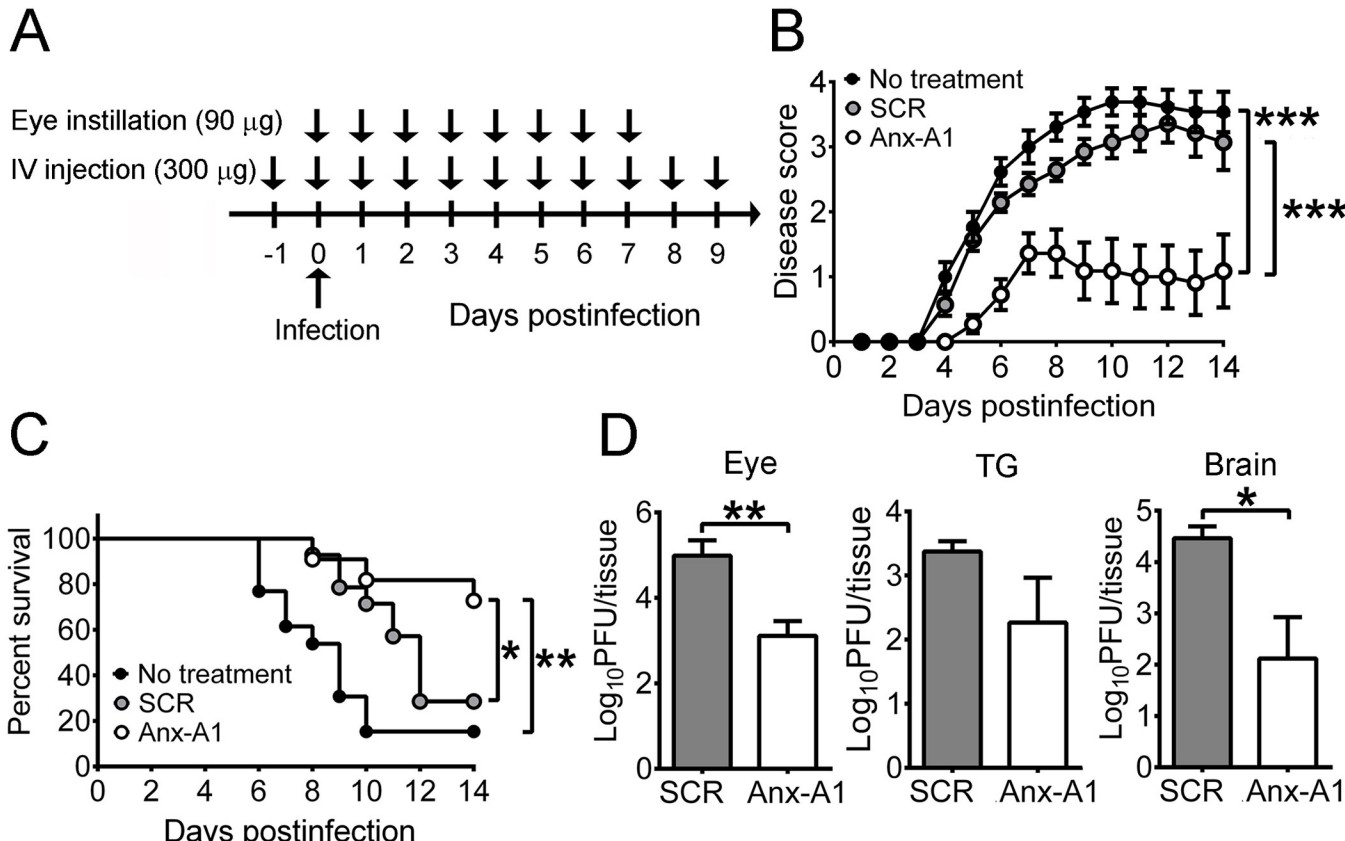

**Fig 6. Suppression of Anx-A1 expression reduces the mortality and tissue viral loads of HSV-1-infected mice.** (**A**) WT mice were treated without or with ODNs on the eye and by intravenous (IV) injection and infected with HSV-1 as indicated. The disease scores (**B**) and survival rates (**C**) of infected mice treated without ODN (No treatment; $n = 13$) or with scramble ODN (SCR; $n = 14$) or *Anx-A1* antisense ODN ($n = 11$) are shown. (**D**) Viral loads in the indicated tissues of mice treated with scramble (SCR; $n = 6$/group) or *Anx-A1* antisense ODN ($n = 7$/group) were determined at 7 dpi. Data show the mean + SEM in panels B and D. $^*P < 0.05$, $^{**}P < 0.01$, and $^{***}P < 0.001$.

The antisense oligodeoxynucleotides (ODN) specifically against *Anx-A1* has been shown to effectively suppress Anx-A1 protein expression in vitro [24,25]. We investigated whether Anx-A1 knockdown can protect mice from HSV-1 infection. WT mice were treated with ODN topically on the eye starting right before infection and systemically by intravenous injection starting one day before infection (Fig 6A). Treatment with *Anx-A1* ODN reduced the Anx-A1 level in the brain of infected mice by 50% when compared with the scramble ODN (S8A Fig). Fig 6B and 6C show that *Anx-A1* ODN treatment reduced the disease severity and increased the survival rate of infected mice (8/11) when compared with the scramble ODN group (4/14) or the group without ODN treatment (2/13). Additionally, *Anx-A1* ODN treatment significantly decreased the viral loads in mouse eyes and brains at 7 dpi (Fig 6D). A lower (one-third) dose of *Anx-A1* ODN failed to increase the survival rate of infected mice (S8B Fig), suggesting that *Anx-A1* ODN protects mice from HSV-1 infection in a dose-dependent manner.

### Blocking FPR2 reduces HSV-1 binding to the cell surface and the mortality as well as tissue viral loads of infected mice

FPR1 and FPR2 are receptors of extracellular Anx-A1 [26,27]. FPR2 was detected on the surface of both A549 and N18 cells with a higher level found on N18 cells than A549 cells (Fig

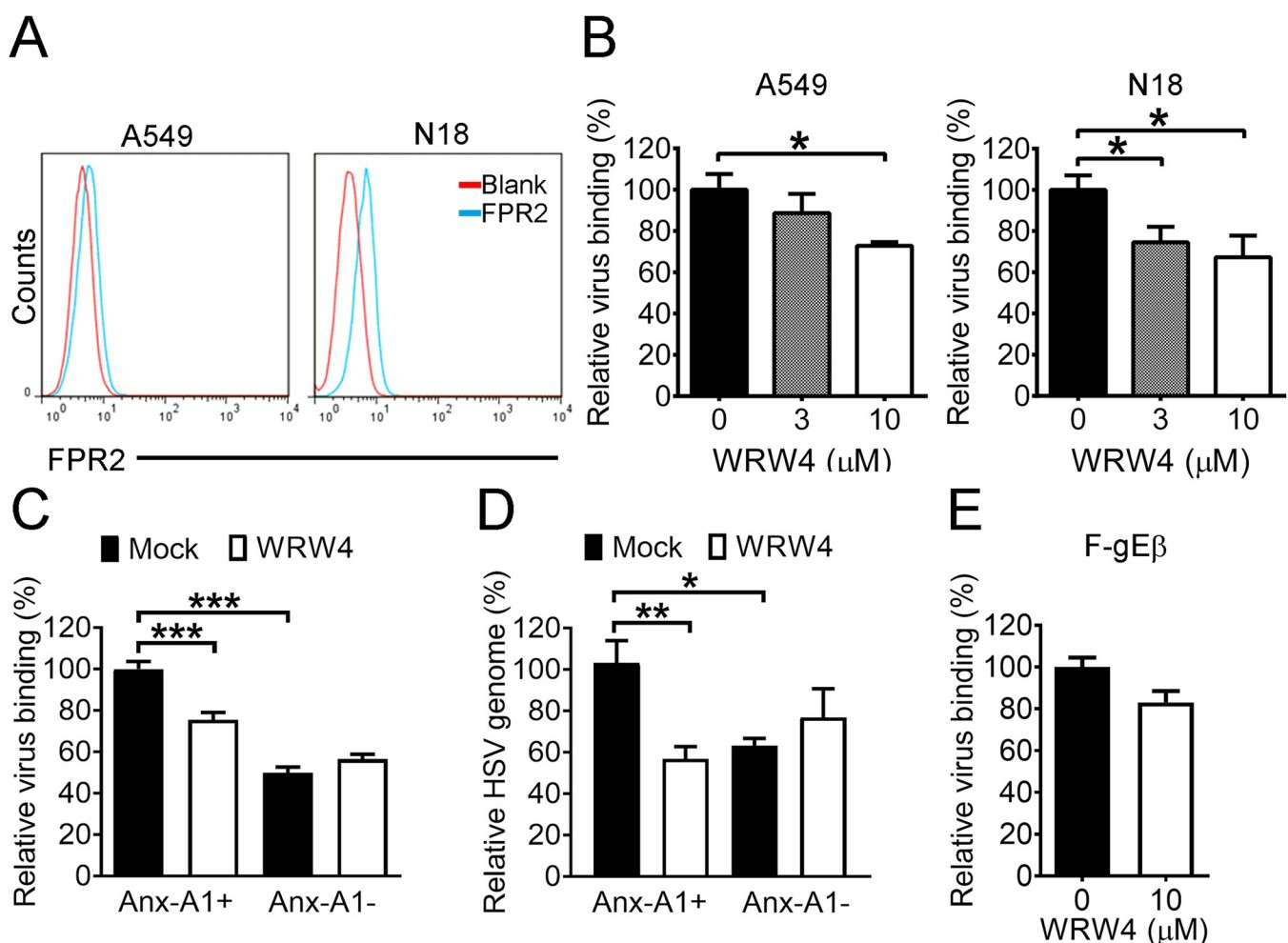

**Fig 7. Blocking FPR2 reduces HSV-1 binding.** (**A**) Representative histograms of FPR2 detected on the surface of the indicated cells are shown. (**B**) The levels of virus binding to the indicated cells treated with the indicated concentrations of WRW4 are shown. The levels of Anx-A1-positive (Anx-A1+) or Anx-A1-negative (Anx-A1-) HSV-1 binding to N18 treated without or with WRW4 (10 μM) were determined by plaque assay (**C**) and real-time PCR (**D**). (**E**) The levels of strain F-gEβ binding to N18 cells treated without or with WRW4 (10 μM) are shown. Panels B and E were determined by plaque assay. The levels of Anx-A1+ virus binding to cells without WRW4 treatment were set as 100% in panels C and D, and the level of strain F-gEβ binding to N18 cells treated without WRW4 was set as 100% in panel E. Data show the mean + SEM of >4 samples per group. $^*P < 0.05$, $^{**}P < 0.01$, and $^{***}P < 0.001$.

7A). The antagonist (WRW4) suppressing the binding of Anx-A1 to FPR2 is a small peptide of six amino acids (Trp-Arg-Trp-Trp-Trp-Trp) [28]. WRW4 treatment reduced the levels of HSV-1 binding to A549 and N18 cells (Fig 7B). Similar results were observed by assessing HSV-1 binding on N18 cells via plaque assay and real-time PCR (Fig 7C and 7D), showing that blocking FPR2 inhibits virus binding to the cell surface. Notably, absence of Anx-A1 (Anx-A1-) or gE (F-gEβ) on HSV-1 virions abrogated the suppressive function of WRW4 on HSV-1 binding (Fig 7C–7E). FPR1 was also detected on the surface of N18 and A549 cells (S9A Fig), but treatment with the FPR1 antagonist [28], Boc-1, failed to affect HSV-1 binding to both N18 and A549 cells (S6B Fig). To test the effect of WRW4 in vivo, we treated WT mice with or without WRW4 both topically and systemically before infection (Fig 8A). The survival rates of infected mice treated with or without WRW4 were 75% (6/8) and 25% (2/8), respectively (Fig 8B). WRW4 treatment significantly decreased the viral load in mouse brains by 10-fold at 5 dpi (Fig 8C).

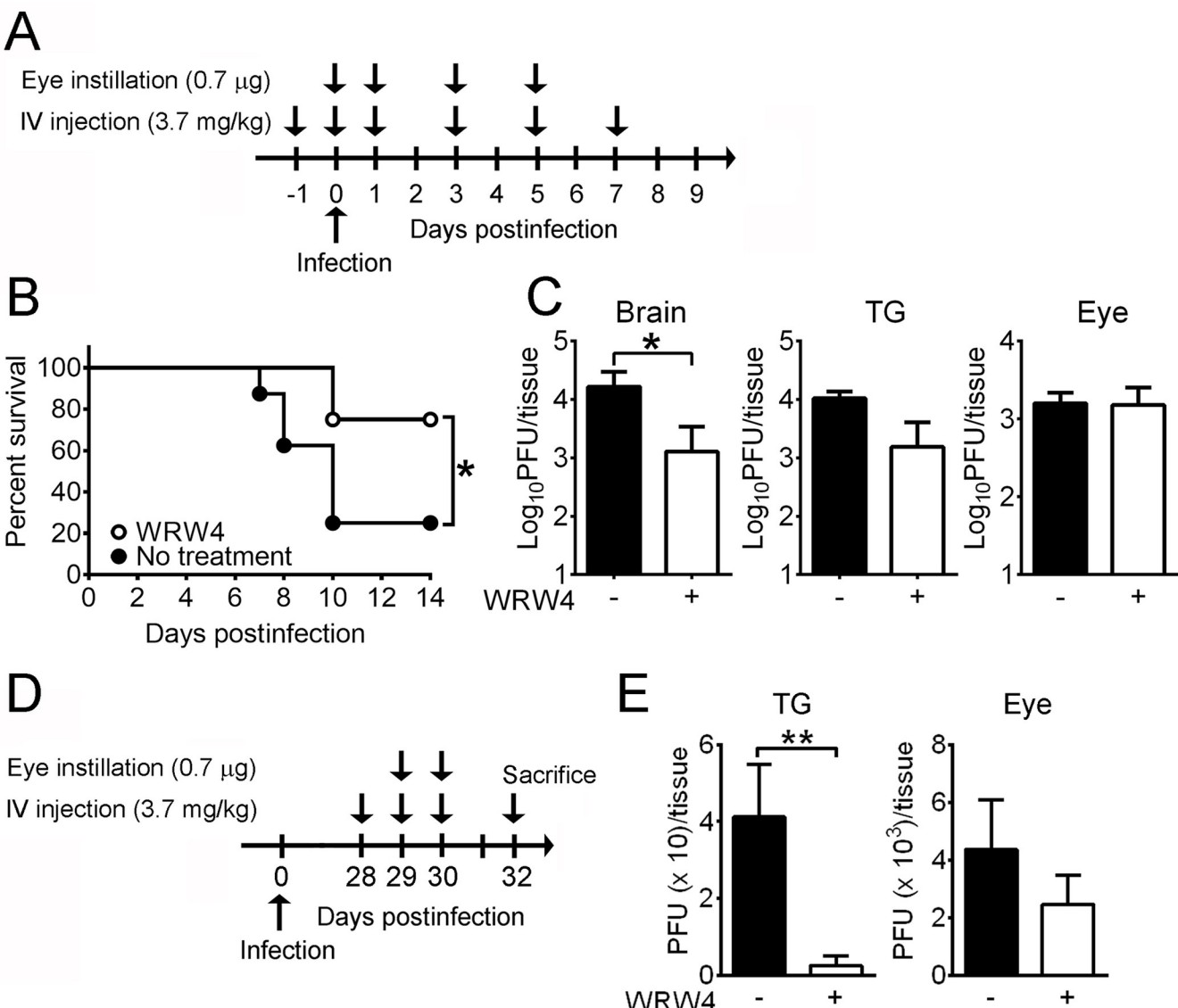

**Fig 8. Blocking FPR2 reduces the the mortality rate and viral load in neural tissues of infected mice.** (**A**) WT mice were treated without or with WRW4 on the eye and by intravenous (IV) injection and infected with HSV-1 as indicated. (**B**) The survival rates of infected mice treated without (No treatment; *n* = 8) or with WRW4 (WRW4; *n* = 8) are shown. (**C**) Viral loads in the indicated tissues of mice treated without (-) or with (+) WRW4 were determined at 5 dpi (*n* = 5/group). (**D**) BALB/c nude mice were infected with HSV-1 294*dl*TKA, treated without (-) or with (+) WRW4 on the eye and by IV injection, and sacrificed as indicated. (**E**) Viral loads in the indicated tissues of mice treated without or with WRW4 were determined (*n* = 8/group). Data show the mean + SEM in panels C and E. $^*P < 0.05$ and $^{**}P < 0.01$.

We further investigated the therapeutic efficacy of WRW4 against acyclovir-resistant HSV-1, which causes serious problems in immunocompromised patients with tendency to develop disseminated and chronic infection [29]. Previously, we infected BALB/c nude mice with the acyclovir-resistant HSV-1, 294*dl*TKA, which establishes persistent infection in the mouse eye and TG [30]. The infected BALB/c nude mice were treated with or without WRW4 one time per day both topically and systemically starting from 28 dpi for 2 and 3 days, respectively (Fig 8D). Without WRW4 treatment, HSV-1 was detected in 87.5% (7/8) of mouse eyes and TG with average titers of 4363 and 41 plaque forming units (PFU)/tissue, respectively (Fig 8E). With WRW4 treatment, HSV-1 was detected in 80% (8/10) of eyes and 12.5% (1/8) of TG with

average titers of 2470 and 3 PFU/tissue, respectively. No infectious virus was detected in the brain or peripheral tissues of infected mice. These results show that WRW4 treatment reduces the viral loads in infected tissues and the percentage of infected TG of BALB/c nude mice with persistent 294*dl*TKA infection.

## Discussion

The ligand of HSV-1 gE remains elusive. Here, we identify Anx-A1 to be the ligand of gE, as gE associates with Anx-A1 in the absence of other viral proteins. Anx-A1 is detected on the cells permissive for HSV-1 before infection and on the virus envelope to enhance virus binding to cells and increase virus lethality and tissue viral loads in mice (Fig 9). More importantly, treatments with the *Anx-A1* ODN or FPR2 antagonist reduce the mortality and tissue viral loads of infected mice.

Our in vitro studies detected Anx-A1 on the surface of uninfected cells and on virus envelopes. Both extracellular Anx-A1 on the cell surface and virus envelopes was required for Anx-A1-enhanced HSV-1 attachment. Hence, absence or neutralization of extracellular Anx-A1 reduced HSV-1 attachment to cells (Fig 9). These results are consistent with previous studies showing that Anx-A1 binds membranes of vesicles and cells to promote liposome and exosome aggregation [31,32] and myoblast fusion during myotube differentiation and regeneration [33,34]. Neutralizing Anx-A1 via Ab blocks the exosome aggregation and myotube

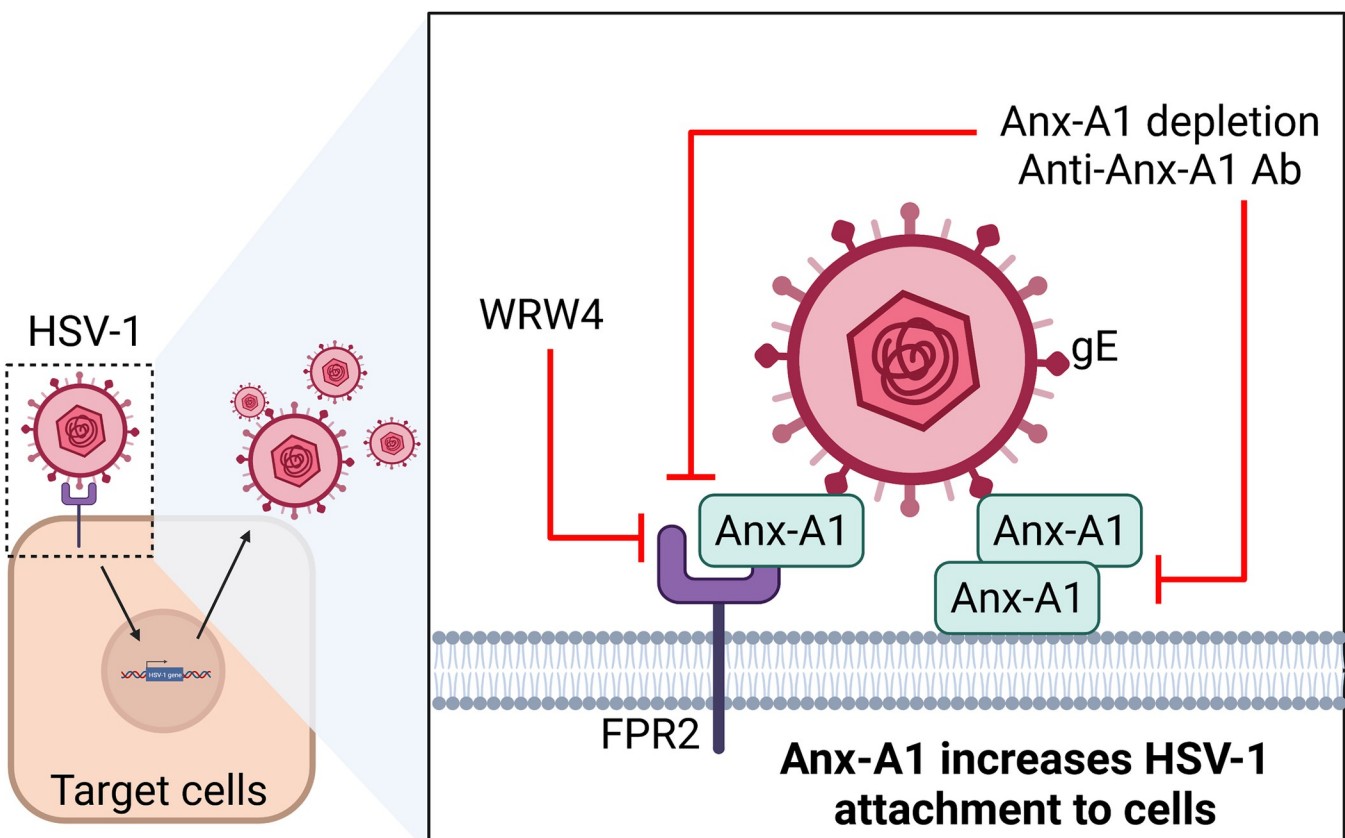

**Fig 9. The schematic diagram of Anx-A1-enhanced HSV-1 attachment to cells.** Anx-A1 binds HSV-1 gE, and Anx-A1 on virions enhances HSV-1 attachment in a versatile manner. Virion Anx-A1 either interacts with Anx-A1 on the cell surface or with FPR2. Depletion of extracellular Anx-A1 or blocking FPR2 via WRW4 inhibits HSV-1 attachment to cells. Created with BioRender.com.

differentiation [32,34]. Extracellular Anx-A1 is proposed to promote membrane aggregation in two ways. Anx-A1 links two juxtaposed membranes by forming homodimers. Alternatively, the N-terminus of Anx-A1 serves as a secondary membrane binding site [31]. Results of Fig 4D suggest that Anx-A1 forms homodimers to facilitate HSV-1 attachment, as the absence of Anx-A1 on either the cell surface or virus envelopes impaired HSV-1 attachment to cells. Moreover, Anx-A1 on virions promotes HSV-1 attachment to FPR2, but not FPR1, showing that Anx-A1 on virus envelopes provides a versatile way for HSV-1 to attach on the cell surface.

Anx-A1 might be associated with HSV-1 envelopes via gE or negatively charged phospholipids. Our results suggest that Anx-A1 binds HSV-1 via gE rather than negatively charged phospholipids since absence of gE on HSV-1 virions abrogated Anx-A1-enhanced binding to MEFs, and the inhibition effect of WRW4 was abrogated in cells infected with gE-null HSV-1. Notably, the level of strain F binding to $Anx-A1^{-/-}$ MEFs was higher than that of strain F-gEβ to WT MEFs, suggesting that gE might bind other cellular proteins, which remain to be identified. Dingwell *et al.* characterized the function of gE by constructing the mutant virus (F-gEβ), which displays a small plaque phenotype [9]. They found that the penetration of F-gEβ into cells was decreased by approximately 25% when compared with that of WT virus, after incubation of infected cells at 37°C for 0.5 h and subsequent treatment with citrate buffer to release or inactivate unpenetrated viruses [9]. The authors further demonstrated that gE and gI form a complex to facilitate virus spread, especially between cells with extensive junctions [9,21,35]. Using F-gEβ, we showed that the levels of HSV-1 binding to cells were dependent on both Anx-A1 and gE. In addition to the different assays that were used in the studies by Dingwell *et al.* [9] and ours, the differences between the two studies might be due to the MOI used to infect cells. We used a MOI of 1 to perform the binding assay. When the MOI was reduced to 0.1, the levels of HSV-1 binding to WT (99.8 ± 31%) or $Anx-A1^{-/-}$ MEFs (67.9 ± 22%) were comparable. Our result highlighting the importance of gE in HSV-1 infection is consistent with the finding that anti-gE Ab neutralizes viral infectivity [10]. Furthermore, gE deficiency reduces HSV-1 neurovirulence in rabbits and mice with decreases of brain viral loads [8,10,36] in manners similar to our finding that suppression of Anx-A1 or FPR2 reduces HSV-1 neurovirulence and brain virus titers in mice. HSV-1 gE is known to facilitate the cell-to-cell spread of HSV-1 [9,21,35], which is mediated by syncytium formation and by directing virus egress at the intercellular spaces [37]. Interestingly, Anx-A1 is shown to promote syncytium formation in reovirus- or measles virus-infected cells [38]. However, the previous study using HSV-1 strain KOS [39] and our study using strains KOS and 294.1 found that both KOS and 294.1 fail to induce syncytium formation in MEFs and all cell lines tested in our study. Further studies are needed to clarify the role of Anx-A1-gE interaction in promoting syncytium formation with the HSV-1 strain capable of inducing syncytium. For directing HSV-1 egress and gE to the intercellular space, the gE cytoplasmic tail is required [40,41]. However, the gE cytoplasmic tail is dispensable for Anx-A1 binding (Fig 3D), suggesting that Anx-A1-gE interaction does not direct HSV-1 egress at the intercellular space. As Anx-A1 binds gE extracellular domain, we focused on Anx-A1 in promoting HSV-1 infection at the initial phase (attachment). We showed that blocking Anx-A1 or FPR2 decreases HSV-1 attachment to cells by 20 to 30% and ameliorates HSV-1-induced encephalitis in mice.

In addition to HSV-1, Anx-A1 is shown to mediate the infection of other viruses, such as cytomegalovirus [19]. Anx-A1 has been detected on the envelope of influenza virus, but the issue regarding whether Anx-A1 binds to the viral (envelope) protein remains elusive [42]. Very few studies have investigated the role of Anx-A1 in virus infections in vivo. Here, we show that Anx-A1 deficiency decreases HSV-1 lethality in mice by 50%, a rate higher than those (approximately 20–40%) mediated by nectin-1 and non-muscle myosin IIA, the receptors of the essential envelope glycoproteins, gD and gB, respectively [13,43]. Our additional results

demonstrated that Anx-A1 also aggravated HSV-2 lethality in WT mice when compared with *Anx-A1*$^{-/-}$ mice, demonstrated by the survival rates of 16.7% (2/12) in WT and 72.7% (8/11) in *Anx-A1*$^{-/-}$ mice, respectively and elevated tissue viral loads, especially in the TG and brains. As Anx-A1 deficiency protects mice from HSV-1 infection, we determined and found that pretreatments with *Anx-A1* ODN and WRW4 topically and systemically diminished HSV-1 lethality and tissue viral loads in mice. Treatments with *Anx-A1* ODN and WRW4 only topically, systemically, or after infection failed to significantly increase the survival rates of infected mice. More importantly, posttreatment with WRW4 significantly suppressed the tissue viral load of BALB/c nude mice persistently infected with acyclovir-resistant 294*dl*TKA. Therefore, future anti-HSV-1 treatments designed to target Anx-A1 can be considered for prophylaxis or in combination with other therapies, such as acyclovir and its derivatives. We also tested anti-Anx-A1 Ab to reduce HSV-1 lethality in mice, but the treatment failed. Anx-A1 is expressed on neutrophils, monocytes, macrophages, and T cells, which protect mice from HSV-1 infection [44–46]. Anti-Anx-A1 Ab treatment may deplete these protective leukocytes and result in the failure to reduce HSV-1 infection. Unlike anti-Anx-A1 Ab, treatments with *Anx-A1* ODN and WRW4 may enhance host immunity, as Anx-A1 and FPR2 have been shown to inhibit host immune responses [47].

We assessed HSV-1 production in MEFs and found that WT MEFs consistently produced more virus than *Anx-A1*$^{-/-}$ MEFs. We noticed that different batches of WT MEFs varied in their total Anx-A1 levels. Moreover, the viral titers and total Anx-A1 levels in MEFs were positively correlated. In the future, it will be of interest to investigate the correlation between Anx-A1 levels and human susceptibility to HSV-1 encephalitis. Collectively, our work provides information on Anx-A1 in HSV-1 pathogenesis and identifies Anx-A1 and its receptor (FPR2) as potential cellular targets for reducing virus-induced lethality and markers for screening human susceptibility to HSV-1 encephalitis.

## Materials and methods

### Ethics statement

Wild-type (WT) C57BL/6 mice (The Jackson Laboratory), C57BL/6-derived *Anx-A1* knockout (*Anx-A1*$^{-/-}$) mice, kindly provided by Roderick Flower [17], and BALB/c nude mice (Bio-LASCO Taiwan Co., Ltd.) were maintained under specific-pathogen-free conditions in the Laboratory Animal Center of our university. The care and use of mice protocols were approved by the Institutional Animal Care and Use Committee in National Cheng Kung University with the approval numbers of 104283 and 106226.

### Cells and viruses

Human cell lines (A549, 293, and U-2 OS), the mouse cell line (N18), and Vero cells were maintained according to the instructions of the American Type Culture Collection. Primary MEFs were cultured from mice and maintained as previously described [48]. HSV-1 strains KOS, KOS-GFP [49], 294.1, 294*dl*TKA, F, and F-gEβ were propagated and titrated on Vero cell monolayers. HSV-1 with or without Anx-A1 was generated by propagating the virus in WT or *Anx-A1*$^{-/-}$ MEFs, respectively. Recombinant adenovirus expressing HSV-1 gE, kindly provided by David Johnson [11], was propagated and titrated on 293 cell monolayers. For most in vitro studies, cells were infected with 294.1 at the MOI of 1 unless specified.

### Immunofluorescence staining and flow cytometry

For staining of Anx-A1 on the cell surface, mock-infected A549 cells were stained with rabbit anti-Anx-A1 serum at 4°C for 2 h before fixation and permeabilization. Alexa 488-conjugated

secondary Ab (Invitrogen) was used to visualize signals in fixed cells and sections under a fluorescence microscope. Nuclei were stained with Hoechst 33258. Cells were treated with control or anti-Anx-A1 serum, infected with KOS-GFP (MOI = 10), washed, and incubated for 16 h. The mock-infected cells were trypsinized and stained with the FPR2 Ab (clone 304405, R&D Systems) conjugated with fluorescein to detect FPR2 on the cell surface. The fluorescent signal on cells was detected by FACSCalibur and analyzed by the FlowJo software.

### Plasmids and generation of rabbit anti-Anx-A1 serum

The *Anx-A1* cDNA was amplified from plasmid pUNO1-Anx-A1 (InvivoGen) by PCR with primers (F 5'-GGAATTCCATATGGCAATGGTATCAGAATTC-3' and R 5'-CCGCTCGAG GTTTCCTCCACAAAGAGC-3'). PCR products were digested with *Nde*I and *Xho*I and cloned into plasmid pET-30a to express recombinant Anx-A1 with a 6×His tag fused at the C-terminus. Recombinant Anx-A1 protein was used to immunize rabbits to generate antiserum (Genetex), which was used for most of the studies in addition to the Anx-A1 Ab purchased from Invitrogen. The *Anx-A1* cDNA was amplified by PCR to generate products encoding full-length and Δ1–42 Anx-A1 with primer pairs 1 (1F 5'- ATCTAGCTCGAGATG GCAATGGTATCAGAATTC-3' and 1R 5'- ATCTAGTCTAGAGTTTCCTCCACAAAGA GCC-3') and 2 (2F 5'-ATCTAGCTCGAGATGTTCAATCCATCCTCGGAT-3' and 2R 5'- ATCTAGTCTAGAGTTTCCTCCACAAAGAGCC-3'), respectively. PCR products were digested with *Xho*I and *Xba*I and cloned into plasmid pCDNA6/myc-His to express recombinant Anx-A1 with a 6×His tag fused at the C-terminus. The DNA encoding full-length, Δ447–550, and Δ1–421 gE were amplified from KOS by PCR with primers pairs A (AF 5'-ATCTAG GAATTCAATGGATCGCGGGGGCG-3' and AR5'- ATCTAGTCTAGACCAGAAGACGGAC GAATC-3'), B (BF 5'-ATCTAGGAATTCAATGGATCGCGGGGGCG-3' and BR 5'-ATCTAG TCTAGACCTGCGCCAACAGGTCAT-3'), and C (CF 5'-ATCTAGGAATTCAATGGCG

GTGATG-3' and CR 5'-ATCTAGTCTAGACCAGAAGACGGACGAATC-3'), respectively. All PCR products were digested with *Eco*RI and *Xba*I and cloned into plasmid p3×Flag-CMV-14 to express gE with 3×Flag fused at the C-terminus. Each individual plasmid was transfected into cells by Lipofectamine 3000 (Thermo Fisher Scientific).

### Assays of virus yield, binding, and penetration

Virus yields were assessed by plaque assay. To reconstitute Anx-A1 expression in MEFs, *Anx-A1$^{-/-}$* MEFs were transfected with an empty vector or the vector expressing recombinant Anx-A1 with a 6×His tag. For assay of virus binding, cells infected with HSV-1 for one h at 4°C, washed at 4°C, immediately stored at -80°C, and subjected to plaque assay as described [20] or to real-time PCR for HSV-1 genomes with primers of *tk*-F: CTTAACAGCGTCAACAGC GTGCCG; *tk*-R: CCAAAGAGGTGCGGGAGTTT; *Adipsin*-F: AGTGTGCGGGGATGCAGT; *Adipsin*-R: ACGCGAGAGCCCCACGTA). The relative HSV-1 genome binds on cell surface was calculated as $\Delta C_T = (C_T tk - C_T Adipsin)$ and $\Delta\Delta C_T = \Delta C_T Anx\text{-}A1^{-/-} - \Delta C_T WT$. For HSV-1 penetration assay, MEF monolayers were infected with HSV-1 at room temperature for 1 h, washed with acidic citric buffer (pH = 3) to inactivate the virus remaining on cell surface, and overlayed with medium containing methylcellulose for plaque formation.

### Virus overlay protein binding assay

HSV-1 virions were purified as described [50]. Briefly, virus-containing medium was centrifuged at 2,020 × g for 10 minutes (min) and 10,000 × g for 30 min at 4°C and loaded on a discontinuous sucrose gradient. Virions were collected at the interphase of the 30–60%.

Recombinant His-tagged Anx-A1 and BSA were loaded onto denaturing SDS-PAGE, separated, transferred onto a membrane, and hybridized with purified virions in protein-binding buffer (100 mM NaCl, 20 mM Tris at pH 7.6, 0.5 mM EDTA, 10% glycerol, 0.1% Tween-20, 2% skim milk, and 1 mM DTT) for 17 h at 4°C. The presence of HSV-1 virions, Anx-A1, and BSA on the membrane was detected by western blotting.

## Co-immunoprecipitation assay

A549 cells were infected with HSV-1 (MOI = 10) for 17 h. Additionally, 293 cells were infected with adenovirus expressing gE at a tissue culture infective dose of 50 for 48 h or transfected with plasmids for 48 h and then mock-infected or infected with HSV-1 (MOI = 10) for 17 h. Cells were lysed in immunoprecipitation buffer (100 mM Tris-HCl, 100 mM NaCl, 0.5% Triton X-100, and 0.5 mM $CaCl_2$) supplemented with protease inhibitors. Lysates were incubated with control Abs or Abs against Anx-A1, gE (clone 9H3, Abcam), or 6×His (Genetex) at 4°C for 17 h and then with magnetic beads coated with protein A/G (Thermo Fisher Scientific) for 1 h. The beads were washed, and the bound proteins were eluted and subjected to western blot analysis.

## Western blot analysis

Cells were lysed with buffer (10 mM Tris-HCl at pH 7.5, 150 mM NaCl, 5 mM EDTA, 5 mM $NaN_3$, 10 mM sodium pyrophosphate, and 1% Triton X-100) supplemented with protease inhibitors. Cell lysates were subjected to western blotting with Abs against Anx-A1, β-actin (clone AC-15, Sigma-Aldrich), BSA (Genetex), gE (clone 3114, kindly provided by David Johnson), gB (clone H1817, EastCoast Bio), gD (clone DL6, Santa Cruz Biotechnology), Flag (clone M2, Sigma), or 6×His (Genetex) and HRP-conjugated secondary Abs. Protein bands were detected by using an enhanced chemiluminescence substrate kit (Millipore). The intensity of the protein bands was measured by ImageJ software.

## Immunogold labeling and transmission electron microscopy

The immunogold labeling procedure was performed on whole or ultrathin-sectioned HSV-1 virions. For immunogold labeling of whole virions, HSV-1 was adsorbed onto a glow-discharged nickel grid (EMS CF-200-Ni) for 10 min and incubated with blocking buffer (1% BSA/phosphate buffered saline) for 1 h before Abs against Anx-A1 or gE were added. One h later, the grid was washed with blocking buffer, incubated with 18 and 10 nm nanogold-conjugated IgG for detecting Anx-A1 and gE, respectively for 1 h, and followed by 1% glutaraldehyde fixation. Finally, the sample was stained with 2% uranyl acetate and air-dried. For immunogold labeling of ultrathin-sectioned HSV-1, sections on a low-discharged nickel grid were treated with 10% $H_2O_2$ for 10 min, blocked for 1 h, treated with mixed anti-AnxA1 and anti-gE Abs for 1 h, incubated with mixed nanogold-conjugated IgG for detecting Anx-A1 and gE for 1 h, and stained with 2% uranyl acetate and 0.25% lead citrate. The sample was inspected by the JEM1400 transmission electron microscope at the magnification of 60,000×, and the images were taken by the 4k × 4k Gatan 895 CCD camera.

## Virus stripping assay

The virus envelope was separated from the tegument and nucleocapsid as described [51]. Briefly, purified HSV-1 virions were resuspended in a buffer (10 mM Tris-HCl at pH 7.5, 1 mM EDTA, 100 mM NaCl, and 1% NP-40) for 15 min and centrifuged at 2,300 × g for 5 min

at 4˚C. The resulting supernatant and pellet were collected as the envelope and tegument plus capsid fractions, respectively.

## Mouse infection study

Six- to eight-week-old WT and *Anx-A1^{-/-}* mice were anesthetized and infected with $2–8 \times 10^6$ PFU/mouse of 294.1 or mock-infected with lysates of uninfected Vero cells topically on the right eye following scarification of the cornea with a needle 20 times. The scramble ODN (5'-G (ps)A(ps)GCCAGCATCTCC(ps)T(ps)T-3') and anti-Anx-A1 (5'-C(ps)C(ps)AGGACCACC TTT(ps)G(ps)T-3') ODN were designed as previously described [24,25] and synthesized by TriLink BioTechnologies with phosphorothioate (ps) modification [52]. Scarified mouse corneas were treated with 5 μl of solution containing ODN and 1 μl of the transfection reagent, FuGene6 (Roche), once daily for 8 days starting from 15 min before infection. Mice were also given one intravenous injection of 200 μl of solution containing ODN and 1.5 μl of FuGene6 once daily for 11 days starting from 1 day before infection. WRW4 was dissolved in DMSO to make a stock of 100 mM and then diluted in saline. Scarified mouse corneas were treated with 2 μl solution containing WRW4 once daily 15 min before infection and at 1, 3, and 5 dpi. Mice were also given one intravenous injection of WRW4 once daily at -1, 0, 1, 3, 5, and 7 dpi. The disease progression and survival of infected mice was monitored. The signs of encephalitis of infected mice were graded as previously described [53] with modifications. Briefly, mice were graded as 0, normal; 1, jumpy; 2, ataxia; 3, hunched back and lethargy; 4, moribund or death. In separate experiments, infected mice were sacrificed, and their tissues were harvested to determine viral loads or for immunofluorescence staining. For BALB/c nude mice, mice were infected $1 \times 10^7$ PFU of 294*dl*TKA in both eyes following scarification of the cornea with a needle 20 times. Infected mice were treated with 2 μl solution containing WRW4 on the mouse corneas and an intravenous injection of WRW4 at 28, 29, and 30 dpi and sacrificed at 32 dpi to harvest mouse tissues. Mouse tissues were homogenized, and the resulting supernatants were collected to titrate the virus by plaque assay.

## Statistics

Data are expressed as the mean ± or + SEM. For statistical comparisons, the levels of viral loads in cells and tissues were analyzed by the Mann-Whitney *U* test, Kaplan-Meier survival curves were analyzed by the log-rank test, and the other results were analyzed by the Student's *t* test. All *P* values are for two-tailed significance tests. We considered a *P* value of <0.05 as statistically significant.

## Supporting information

**S1 Table. HSV-1 proteins bind to Anx-A1 in infected cells.**
(XLSX)

**S1 File. Supporting materials and methods.**
(DOCX)

**S1 Fig. Anx-A1 can be detected on the surface of uninfected cells.** Representative western blots of Anx-A1 and α-tubulin on the surface and in the cytoplasm of mock-infected A549 cells (**A**) and N18 cells (**B**).
(TIF)

**S2 Fig. Staining performed before permeabilization fails to detect the intracellular protein.** A549 cells were stained with phalloidin conjugated with tetramethylrhodamine to detect F-

actin before and after permeabilization. Nuclei were counterstained with Hoechst.
(TIF)

**S3 Fig. Ectopic Anx-A1 expression in *Anx-A1*$^{-/-}$ MEFs.** The representative western blots of indicated proteins in *Anx-A1*$^{-/-}$ MEFs transfected with 30 or 300 ng/culture of control vector (Vector) or the vector expressing Anx-A1 with His-tag (Anx-A1-His) for 24 hours are shown. The *Anx-A1*$^{-/-}$ MEFs transfected with 300 ng/culture plasmid DNA were subjected to HSV-1 infection shown in Fig 2B.
(TIF)

**S4 Fig. Colocalization of Anx-A1 and HSV-1 on the cell surface.** A549 cells were mock-infected or infected with HSV-1 KOS (MOI = 10) labeled with the lipophilic dye DiD at 4˚C for 1 h before the cell culture temperature was shifted to 37˚C for 5 min to enhance virus binding on the cell surface. Cell-surface Anx-A1 was stained with anti-Anx-A1 Ab, and nuclei were counterstained with Hoechst. Arrows indicate the colocalization of Anx-A1 with HSV-1.
(TIF)

**S5 Fig. Blocking Anx-A1 reduces HSV-1 binding to a human cell line.** The level of virus binding to U-2 OS cells treated with control or anti-Anx-A1 serum and infected with HSV-1 (MOI = 1) are shown. The level of virus binding to control serum-treated cells was set as 100%. Data show the mean + SEM of >4 samples per group. $^{*}P < 0.05$.
(TIF)

**S6 Fig. The expression of Anx-A1 in mouse tissues permissive for HSV-1 replication.** (**A**) Representative western blots of Anx-A1 and β-actin in the indicated tissues of mock-infected WT mice are shown in the top panel, and the quantitative result is shown in the bottom panel. The mean value of brain samples was set as 1. Data show the mean + SEM of 3 samples per group. (**B**) The eyes of uninfected WT and *Anx-A1*$^{-/-}$ mice were sectioned and stained with Abs against Anx-A1 or keratin K3. (**C**) The eyes of WT mice mock-infected or infected with HSV-1 294.1 for 1 day were sectioned and stained with Abs against Anx-A1 or HSV-1. E, epithelium. S, stroma. Nuclei were counterstained with Hoechst. Images are representative of at least 3 samples per group from 2 independent experiments.
(TIF)

**S7 Fig. Absence of Anx-A1 reduces HSV-1-infected neurons in mice.** The representative images of mouse brains harvested at 7 dpi, sectioned, and stained with antibodies against NeuN or HSV-1 are shown.
(TIF)

**S8 Fig. *Anx-A1* ODN reduces Anx-A1 expression in the brain of infected mice.** (**A**) Representative western blots (upper panel) and quantitative results (lower panel) of Anx-A1 expressed in the brains of mice treated with scramble (SCR) or *Anx-A1* antisense ODN (shown in Fig 6A) and infected with virus for 7 days are shown. Data show the mean + SEM of 3 samples per group. The relative level of Anx-A1 in the brain of SCR ODN-treated mice was set as 1. (**B**) The survival rates of infected mice treated without ODN (No treatment; *n* = 16) or with reduced (one-third) amounts of scramble ODN (SCR; *n* = 12) or Anx-A1 ODN (*n* = 10) shown in Fig 6A, are shown.
(TIF)

**S9 Fig. Boc-1, an FPR1 antagonist, fails to reduce HSV-1 binding to the cell surface.** (**A**) Representative histograms of FPR1 detected on the surface of the indicated cells are shown. (**B**) The levels of virus binding to cells treated with the indicated concentrations of Boc-1 are

shown. The level of virus binding on cells without Boc-1 treatment was set as 100%. Data show the mean + SEM of 6 samples per group.
(TIF)

## Acknowledgments

We thank Donald Coen and Robert Lausch for helpful suggestions, Roderick Flower and David Johnson for providing essential materials, and Shan-Lin Hung for technical support.

## Author Contributions

**Data curation:** Hui-Wen Yao, Yen-Chi Chiu.

**Funding acquisition:** Li-Chiu Wang, Shun-Hua Chen.

**Investigation:** Li-Chiu Wang, Shang-Rung Wu, Hui-Wen Yao, Pin Ling, Guey-Chuen Perng, Sheng-Min Hsu, Shun-Hua Chen.

**Project administration:** Shun-Hua Chen.

**Resources:** Guey-Chuen Perng.

**Supervision:** Pin Ling, Guey-Chuen Perng, Sheng-Min Hsu, Shun-Hua Chen.

**Writing – original draft:** Li-Chiu Wang, Shang-Rung Wu, Sheng-Min Hsu, Shun-Hua Chen.

**Writing – review & editing:** Li-Chiu Wang, Shang-Rung Wu, Shun-Hua Chen.

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
