## [Decision Letter · Decision Letter 0]

19 Jan 2022

Dear Dr. Chen,

Thank you very much for submitting your manuscript "Suppression of annexin A1 and its receptor reduces herpes simplex virus 1 lethality in mice" for consideration at PLOS Pathogens. As with all papers reviewed by the journal, your manuscript was reviewed by members of the editorial board and by several independent reviewers. In light of the reviews (below this email), we would like to invite the resubmission of a significantly-revised version that takes into account the reviewers' comments.

Please address all the comments of the reviewers. The points raised in the "Major Issues" should be addressed experimentally.

We cannot make any decision about publication until we have seen the revised manuscript and your response to the reviewers' comments. Your revised manuscript is also likely to be sent to reviewers for further evaluation.

Sincerely,

Neal A. DeLuca, Ph.D.

Guest Editor

PLOS Pathogens

Shou-Jiang Gao

Section Editor

PLOS Pathogens

Kasturi Haldar

Editor-in-Chief

PLOS Pathogens

orcid.org/0000-0001-5065-158X

Michael Malim

Editor-in-Chief

PLOS Pathogens

orcid.org/0000-0002-7699-2064

Please address all the comments of the reviewers. The points raised in the "Major Issues" should be addressed experimentally.

Reviewer's Responses to Questions

**Part I - Summary**

Reviewer #1: The authors provide evidence for their claim that HSV-1 glycoprotein gE binds to Anx-A1 on cell surface to promote infection of cells. Their in vivo results show that suppression of Anx-A1 reduces the mortality and viral loads in infected mice. Multiple in vitro assays as well as in vivo assays were performed to validate the claim. The manuscript is well written and the results well described.

Reviewer #2: Wang and colleagues provide evidence that the cell protein annexin A1 (Anx-A1) enhances attachment of HSV-1 to cells via an interaction with gE. Furthermore, that blocking this interaction reduces the lethality of the virus for mice after infection via the cornea.

The experiments with mice consist of comparing wildtype and Anx-A1-negative mice, of treating mice with an oligonucleotide that reduces Anx-A1 expression, and of treating mice with a peptide that blocks binding of the soluble Anx-A1 to its cell surface receptor FPR2. In all cases, the data support a role for Anx-A1.

The results of experiments with tissue culture are less straightforward, in part because some of the methods used are not described adequately. Details are below.

There is novelty and potential significance to the study, and the Discussion is written with appropriate references to prior work.

Comments on the execution and interpretation of the experiments using tissue culture systems are as follows.

(1) In the first experiment, cells were incubated with virus at an MOI of 5 for 1 h at 4C, then incubated at 37C for various times before measurement of cell surface Anx-A1. There was an initial increase, but then the level went back down. The conclusion is that infection raised the level of Anx-A1 on the cell surface, and that UV-inactivated virus had a smaller effect. However, in a later experiment it was shown that Anx-A1 is actually present on virions, which suggests that the increased cell surface level of Anx-A1 was actually due to Anx-A1 on virions. This would be consistent with an initial increase, followed by a decrease, because there is no indication that the inoculum was removed after the 4C step. More virus could have bound after raising the temperature, followed by entry into cells resulting in a later decrease of virions at the cell surface. The UV virus was only tested after 1 h at 37C, at which point the level of Anx-A1 is not greatly different from that seen without the UV treatment.

(2) A virus binding assay is described:

Cells were "infected with HSV-1 for 1 h, washed, and harvested at 4C to determine the number of virus binding to cells by plaque assay."

What is the significance of the plaque assay? This is not usually the way that virus binding is measured, and it is entirely unclear if the procedure is actually measuring binding or a combination of binding and a later step or steps.

(3) In Figs 1C and 1D, immunofluorescence is used to detect increased Anx-A1 at the cell surface after infection. Quantitation shows an approximately 2-fold overall increase. The number of positive cells increases from 32% to 78%. This is sufficient to account for the overall 2-fold increase, so it is difficult to accept the report that fluorescence intensity also increased, i.e., that positive cells became more positive. The images do not support this. And again, is the overall increase the result of Anx-A1 being brought to cells on virions?

(4) In Fig 2A, a virus overlay protein blot assay is used to demonstrate that virions bind to Anx-A1. Details of sample preparation are not given, but presumably this is a denaturing gel, in which case virions are binding to denatured Anx-A1, which is an interesting and surprising observation.

(5) An interaction between gE and Anx-A1 is demonstrated, and said to be independent of other virus proteins (Fig. 2C). Nevertheless, quantitation of the gel would be useful, comparing the fraction of total Anx-A1 that is co-IP'd with gE; is it the same when only gE is present versus when all virus glycoproteins are present?

(6) Anx-A1 interacts with the extracellular domain of gE. The experiment to demonstrate that it does not interact with the cytoplasmic domain (Fig. 2D) suffers from the problem that the hydrophobic transmembrane region is also included in the construct. This would likely result in aggregation of the cytoplasmic domain, preventing any meaningful conclusion.

(7) Figs. 3A and 3C are concerned with the effect of anti-Anx-A1 serum on binding of virus to cells. The effect is measured via plaque assay. As with comment (2), this is unclear. How does a plaque assay measure virus binding? Also, how can antibody binding to the Anx-A1 on virions be eliminated as a major contributor to the result?

(8) Yields of virus are measured in Figs 3B and 3D. Time points are 24 h and 36 h. A greater effect of anti-Anx-A1 antibody is seen at 36 h. The problem with this protocol is that by 36 h, there will have been two cycles of infection. Therefore, the effect of the antibody on virus spread from cell to cell is also a factor. Curiously, in the next experiments, with Anx-A1-negative cells, the results suggest that spread is NOT affected. To eliminate this complication, it would be preferable to use a higher MOI for the initial infection in all of these experiments, and to measure the virus yield at no later than 24 h.

(9) There is an experiment to investigate a role for Anx-A1 in virus entry, i.e., a post-binding step, but the experimental protocol is unusual. Virus was incubated with cells for 1 h at room temperature, followed immediately by a low pH treatment to inactivate virions still on the cell surface, and then incubation at 37C to allow plaque formation. The assay therefore measures a combination of virus binding and entry into cells at room temperature, which is not very informative.

The conventional way to assess entry is to begin with an initial virus attachment step at 4C. Next to raise the temperature to 37C to allow virus entry, with low pH treatment at various times after temperature increase. This allows the rate of virus entry to be determined.

(10) A virus lacking gE is used in Fig 3I. Do the authors have an explanation for why the level of binding (if that is what is being measured) when gE is absent but Anx-A1 is present is only half that of when gE is present but Anx-A1 is absent?

(11) Fig 4 shows that Anx-A1 is present on virions, as mentioned earlier. The statement is also made that Anx-A1 is "adjacent to gE". This statement cannot be justified without comparing the proximity of Anx-A1 to other virus glycoproteins, which was not done.

(12) For Figs 4D-4F, with cells lacking Anx-A1, the level of binding is said to be "slightly higher" for virus containing Anx-A1 than for virus lacking Anx-A1 at an MOI of 1, and with a more pronounced difference at lower MOIs. It is difficult to appreciate how this effect could be influenced by the MOI. Is it caused by the assay measuring something other than, or additional to, binding? Is it influenced by the unavoidable presence of non-infectious virions in the inoculum?

(13) In Fig 7, a peptide that blocks Anx-A1 binding to its cell surface receptor FPR2 reduces virus binding to cells. There are two negative controls that would strengthen the conclusion. One is to show that the peptide has no effect on virus binding to Anx-A1-negative cells. The other is to show that the peptide has no effect on gE-negative virus binding to Anx-A1-positive cells.

**Part II – Major Issues: Key Experiments Required for Acceptance**

Reviewer #1: Anx-A1 is mostly cytosolic. Could it reduce cell-to-cell spread by neutralizing gE intracellularly? Similar things could happen when neighboring cells secrete the protein. This alternate possibility should be analyzed.

Why did the authors choose not to study cell-to-cell spread when gE is a known player there? A simple cell mixing (infected to uninfected) assay can be done to demonstrate effect on spread of fluorescent virions.

Figs. 1/3, Results (and interpretations) are confusing. Anx-A1 expression goes up 1-2h later, by that time most virions should have been internalized. Several experts believe that virus entry happens within 5 min.

Fig. 2. Results shown in this figure are very important. However, it is not clear whether a loading or input protein control was used.

Fig. 4, results are confusing. Normally the presence of gE next to Anx-A1 on viral envelope should reduce the ability of gE to interact with Anx-A1 on the cell membrane.

Fig. 5, Could the results shown here be due to stronger compensatory mechanisms that takeover the normal functions of Anx-A1?

Reviewer #2: Key modifications/new experiments:

(1) Experimental protocols must be changed so that the relevance of Anx-A1 to individual aspects of virus attachment, entry, and virus yield is measured, rather than a combination of them.

Binding should be measured by an assay that does not allow for subsequent entry, and not by plaque assay.

Entry, or more informatively, rate of entry should be measured by an assay that begins with attachment under conditions where entry cannot occur, and then proceeds to different lengths of time at 37C such that entry can occur.

Virus yield should be measured up to, but not exceeding, 24 h post-infection, and with an appropriately high MOI, such that only one cycle of replication is possible. This avoids the complication of effects on virus spread, in addition to attachment and entry.

(2) The effect of Anx-A1 on virions must be addressed. If this is the source of the increase in cell surface Anx-A1 after virus binding/infection, then one conclusion of the manuscript is invalidated. This conclusion is that infection results in exit of intracellular Anx-A1 from cells, such that it is then available to bind to the cell surface.

(3) Negative controls are needed for experiments with FPR2. Does the peptide have an effect on virus binding to Anx-A1-negative cells, and does it have an effect on gE-negative virus binding to Anx-A1-positive cells?

If it does, then a different interpretation of the results already in the manuscript would be needed.

**Part III – Minor Issues: Editorial and Data Presentation Modifications**

Reviewer #1: (No Response)

Reviewer #2: None.

PLOS authors have the option to publish the peer review history of their article (what does this mean?). If published, this will include your full peer review and any attached files.

Reviewer #1: No

Reviewer #2: No
---

## [Editor Report · Decision Letter 1]

20 Jun 2022

Dear Dr. Chen,

We are pleased to inform you that your manuscript 'Suppression of annexin A1 and its receptor reduces herpes simplex virus 1 lethality in mice' has been provisionally accepted for publication in PLOS Pathogens.

Best regards,

Neal A. DeLuca, Ph.D.

Guest Editor

PLOS Pathogens

Shou-Jiang Gao

Section Editor

PLOS Pathogens

Kasturi Haldar

Editor-in-Chief

PLOS Pathogens

orcid.org/0000-0001-5065-158X

Michael Malim

Editor-in-Chief

PLOS Pathogens

orcid.org/0000-0002-7699-2064

All the concerns have been addressed.
---

## [Editor Report · Acceptance letter]

24 Jul 2022

Dear Dr. Chen,

We are delighted to inform you that your manuscript, "Suppression of annexin A1 and its receptor reduces herpes simplex virus 1 lethality in mice," has been formally accepted for publication in PLOS Pathogens.

Best regards,

Kasturi Haldar

Editor-in-Chief

PLOS Pathogens

orcid.org/0000-0001-5065-158X

Michael Malim

Editor-in-Chief

PLOS Pathogens

orcid.org/0000-0002-7699-2064